# Analytical Study Regarding the Behavior of Cr-Co and Ni-Cr in Saliva

**DOI:** 10.3390/medicina58111524

**Published:** 2022-10-26

**Authors:** Adina Oana Armencia, Magda Antohe, Cătălina Iulia Săveanu, Dana Budală, Carina Balcoș, Otilia-Sanda Prelipceanu, Marius Prelipceanu, Dragoș Ionuț Vicoveanu

**Affiliations:** 1Surgery I Department, Faculty of Dental Medicine, “Gr. T. Popa” University of Medicine and Pharmacy, 700115 Iasi, Romania; 2Prosthodontics Department, Faculty of Dental Medicine, “Gr. T. Popa” University of Medicine and Pharmacy, 700115 Iasi, Romania; 3Faculty of Physics, “Al. I. Cuza” University, 700115 Iasi, Romania; 4Department of Computers, Electronics and Automation, Faculty of Electrical Engineering and Computer Science, “Stefan cel Mare” University of Suceava, 720225 Suceava, Romania

**Keywords:** corrosion, dental alloy, saliva

## Abstract

*Background and Objectives*: The interaction between dental alloys and saliva affects both its own properties and those of metallic materials. *Materials and Methods*: Samples made of Cr-Co and Ni-Cr were studied. It was opted for corrosion under tension, scanning electron microscopy was used to characterize the surface morphology, and the chemical composition of the surface was assessed with the help of an Energy Dispersive Spectrometer. In vitro testing of the cytotoxic impact of the study eluates was carried out by flow cytometric analysis. *Results*: Pitting areas appear in the mass of the Ni-Cr alloy. Nickel, Manganese, and Cobalt dissolve and go into the solution. Corrosion is superficial in the case of the Cr-Co alloy, the corrosion points are shallow, and the amount of dissolved metal is relatively small. Mostly Nickel passes into the solution, unlike Chromium and Cobalt, which remain at this level. We noticed an increase in the viability of cell cultures in the case of Cr-Co alloy and a decrease in the number of living cells (87%) for Ni-Cr alloy. *Conclusions*: Common alloys (Ni-Cr and Cr-Co) are prone to corrosion, because they lack structural features that would shield the alloy from corrosion agents.

## 1. Introduction

In the oral environment, restorative materials (metallic or acrylic) must be considered as forming a unitary whole with their organic support. These materials are permanently subject to the cumulative action of mechanical, physicochemical or physical factors that have the effect of changing the structure of the material or even compromising the reliability of the restoration.

An important role belongs to the physicochemical factors, represented, in particular, by the oral fluid such saliva, mucus, gingival fluid, peri-cervical fluid, epithelial cells, scaly cells, etc. Saliva has a high buffering capacity due to the phosphate and carbonate content but also due to the mucin in the composition. Lower pH values can determine, through differentiated aeration, chemical and electrochemical corrosion at the level of metal prosthetic restorations. Therefore, ions are released into the oral environment as a direct result of this event, which has detrimental consequences on both the restoration’s surrounding soft tissues and the restoration itself [1,2].

Even noble metals and metal alloys used in the restoration are prone to corrosion, which is the formation of compounds with environmental variables that have different properties from the basic alloy [3].

The phenomenon of corrosion must be known and combated in order to ensure the reliability of the prosthetic device and to avoid the occurrence of unwanted biological processes. According to Brugirard, three basic corrosion processes—chemical corrosion, electrochemical corrosion, and biological corrosion—can occur in the oral environment. These phenomena are influenced by the chemical components produced in the oral fluid and the heterogeneity of the alloy [1].

Talbot and Lacombe note the existence of three types of metal corrosion: generalized corrosion, pitting (spot corrosion), and crack corrosion (inter-crystalline). Chemical corrosion, and especially electrochemical corrosion, are the two forms of this complex process that occur when a potential difference is established between two conductors by placing them in an electrolyte solution, as in the case of saliva [2].

A large potential difference between two metals or alloys creates a greater tendency for electrochemical corrosion. Thus, the metallic material located below the electrolytic scale constitutes the anode that corrodes by emitting ions. These ions are captured by the cathode, i.e., the metallic material located above on the electrolytic scale. Therefore, the anode corrodes and becomes shiny while the cathode is charged, by electrodeposition, with the ions of the anode, changing its surface condition and mattifying or even coloring it [3,4,5].

Noble alloys have a high corrosion resistance only when the noble metals in the composition represent at least 75%. For Ag-Pd or Au-Pd-Ag alloys, the amount of noble metal must exceed 25% by weight to protect the alloy from the action of corrosive agents. The corrosion resistance of Fe-Cr-Ni or Ni-Cr stainless alloys is good provided that the technological conditions of casting and processing are observed. Ordinary (or common) alloys Fe-Cr-Ni, Cu-Ni, or Al-Cu-Ni are susceptible to corrosion because they do not have elements in the structure that protect the alloy against corrosion agents. High-alloy Ni-Cr steels have high corrosion resistance, and the addition of Molybdenum (Mo) substantially reduces the corrosion rate [6,7,8].

Under the conditions of the oral environment, by modifying the saliva qualities, the restorative materials suffer a series of damages through the corrosion process and the release of some components that modify their qualities. This largely contributes to the modification of the saliva qualities (pH variations).

Any change in the balance of the network in the salivary environment, such as an increase in the concentration of that metal in saliva, leads to significant structural changes and network distortions or even its separation as metal or chemical compounds; the network no longer supports the atoms of that metal, which can be released into the environment, exerting sometimes harmful effects on oral structures [6].

Due to the nature and structure of dental metals, they can corrode chemically, electrochemically, and biologically in the oral cavity. Electrochemical corrosion is the most substantial type of corrosion. Corrosion effects range from alloy color changes to denture cracks or fractures, especially extensions. Pigmentation (sometimes extending to the adjacent mucosa), carious lesions, and even corono-radicular damage occur in the teeth.

Saliva, as an oral biotope, is a continuous environment conducive to corrosion. It disintegrates substances from the outside environment (food, drinks) that alter salivary pH and composition. The content, structure, and properties of the employed alloys, as well as the composition and pH of the saliva, greatly influence the severity and impacts of corrosion. A poor alloy selection may cause failure, either immediately or later. These could have harmful impacts on the patient’s health.

A hypoxic environment may occur because of elements released from alloys, which can disrupt cell physiology and produce cytotoxicity and inflammatory reactions but also areas of necrosis. The nature of the elements released into the oral environment determines the cytotoxic effect of an alloy [6].

The present study analyzes the in vitro interaction between non-noble dental alloys and artificial saliva by focusing on the effect of saliva on alloy corrosion through SEM and EDX and evaluating the cytotoxic impact of the ions released by protein synthesis and flow cytometry.

## 2. Materials and Methods

Metal samples, Cr-Co alloy (Co 59%, Cr 25%, W 10%, Mo 4%, Si 1%, Mn 0.8%), and Cr-Ni alloy (Ni 62.7%, Cr 24.5%, Mo 10%, Si 1.35%, Fe 1.0%), in the form of metal plates with dimensions no greater than 2 cm^2^, were examined in order to accomplish these goals. Following the guidelines of the production businesses, the instruments were used successively for the casting of the alloys, processing, and the finishing of the samples.

The grinding of the surface of the metal alloys to be examined was made with granules of different sizes (10–1200 µ). The surface was finally polished in two stages: at first, we polished the surfaces with diamond paste (1 granules) and, in the last step, with diamond paste with lower granulation (0.25 granules) (Diateh, Bucharest, Romania).

To determine the concentration of ions released, the samples to be examined were introduced into artificial saliva (Duffo-Quezada), whose composition simulates natural saliva at a pH range of 2, 4, 6, and 8, and were subsequently examined microscopically. This mixture was chosen because, according to some research, it exhibits corrosive qualities that are quite like those of normal saliva [9].

We chose live corrosion, keeping the voltage of the working electrode constant at 2 V, when the density of the corrosion current is quite high, to significantly shorten the amount of time needed to assess the interaction between alloys and fake saliva (over 1000 times higher than the density instantaneous current in the absence of a power applied to the electrode).

In every test, the amount of synthetic saliva in the cell was 50 mL, the platinum electrode and alloy sample voltage were 2 V, and the corrosion time was 30 min. The positive pole of the source was attached to the alloy sample.

The appearance of the metal surface was characterized before and after electrochemical corrosion using scanning electron microscopy (SEM), and a Dispersive Energy Spectrometer was used to determine the chemical makeup of the surface. The EDX QUANTAX QX2 detector (Bruker/Roentec Co., Berlin, Germany) was utilized with a VEGA II LSH microscope (Tescan Co., Brno, Czech Republic). The instrument has an electronic cannon with a tungsten filament that allows for a resolution of 3 nm/30 kV acceleration and a magnification power ranging from ×13 to ×1,000,000, and it is totally computer-operated. It works at a pressure of less than 10^−3^ Pa. 

Images were acquired using the VEGA TC Software, and microanalysis was managed using the ESPRIT Software, which supports several analytical techniques, including one-point analysis, one-way analysis, and surface analysis.

The surface microstructures of the alloys before and after corrosion, in both flat and three-dimensional images, as well as the surface compositions before and after treatment, were obtained by EDX microanalysis. All SEM micrographs were taken at the same magnification: ×5000. Elemental microanalysis and EDX spectrum were quantified over an area of 3500 (µm^2^) for all samples.

In the circuit of cytotoxicity tests, the two preparations were coded as follows:SA-Ni-Cr alloy (ionically diffused in artificial saliva)SA-Cr-Co alloy (ionically diffused in artificial saliva)

Kidney cell cultures were obtained from the monkey, Cercopithecus aethiops, cultures that were stabilized, normal, and free of mycoplasma. These cultures were used as biological material for in vitro tests to evaluate the cytotoxic impact of the eluate or, more precisely, of the ionic diffused. These cultures were maintained in plates of 25 cm^2^ that contained DMEM growth medium (Dulbeco’s Modified Essential Medium, Biochrom AG, Berlin, Germany) and were placed in an incubator in a humidified medium, with 5% CO_2_ and a temperature of 370 °C. This was supplemented with 2% fetal bovine serum, 100 g/mL streptomycin (Biochrom AG, Berlin, Germany), 100 IU/mL penicillin (Biochrom AG, Berlin, Germany), and 50 g/mL amphotericin B (Biochrom AG, Berlin, Germany) [10].

Cell cultures that were started in 75 cm^2^ vials to obtain the cell mass required for in vitro research reached the monolayer stage when RM cells confluence. At this point, the cells were separated from the plate’s lower solid substrate using a trypsin solution, 0.25% and 0.02% for EDTA (ethylenediaminetetraacetic acid, Biochrom AG, Berlin, Germany), then resuspended in regular media after being centrifuged at 1800 rpm for 2 min. Test tubes were inoculated with volumes of 2 mL of cell suspension that had a density of 1 × 10^5^ cells per ml and were kept in the incubator at various times under the same environmental conditions [10].

After the monolayer phase of the daughter cell cultures was completed, the growth medium in the test tubes was decanted and replaced with either a standard medium (for control cultures) or one containing the liquid samples to be tested in a volume of 0.2 mL/mL culture medium (in the case of treated cultures) [10].

The growth medium was removed from the 72-h-old cultures after another in vitro developing interval, giving the cells a 48-h treatment. The cell layer was treated with PBS (phosphate-buffered saline), and the total protein content was biochemically assessed using the Lowry method modified by Oyama.

The total protein content of the control and treated cultures (measured in µg protein per culture) was transformed into an index for evaluating the cytotoxic effect of various biopreparations. This is the stage of cell culture development; its disruption is a concretization of their mitosis inhibitory, protein synthesis inhibitory, and disruptive effects on apoptosis and cell viability.

A new experimental model was made to enable the evolution analysis of various variants of RM cell cultures using modern flow cytometry. This was chosen to obtain clear indications of the interaction of the investigated agents with the cell mitosis process, apoptosis, and cell viability.

Thus, 24 h after the establishment of the first cell cultures, 25 cm^2^ culture flasks were divided into two experimental variations, one for analysis of cell proliferation and the other for examining apoptosis and cell viability. Then, 24-h RM cell cultures for flow cytometric analysis of cell division were verified, centrifuged at 1800 rpm/3 min, and resuspended in full growth medium to obtain the cell suspension required for analysis. Following the determination of the number of cells per mL, by classical cytometry with the Turk hemocytometer, the cell suspension was diluted to reach a concentration of 106 cells/mL. The various control and treated control suspensions were preliminarily treated with carboxyfluorescein succinimidyl ester (CFSE) in DMEM culture medium with 10% fetal bovine serum and incubated at 37 °C for 10 min.

Daily, the cells were collected in an enclosure corresponding to a temporary interval in a completed culture medium. They were centrifuged at 1800 rpm/3 min, washed once with PBS, and resuspended in PBS, and the flow cytometry was analyzed with a flow-cytometer with 488 nm excitation, the fluorescence being collected with fluorescein-specific emission filters (FITC).

Covalently attaching to the cytoplasmic elements of the cell, the fluorochrome produced fluorescence uniformly. A blue laser flow-cytometer with a power of 488 nm can detect up to eight cycles of cell division because the dye is uniformly divided across the two daughter cells during cell division, which will emit a more muted fluorescence.

The peak of CFSE-labeled cells was analyzed. The 24-h MRI cultures, both as a control (untreated) and treated as described above, were maintained in the incubator to develop under the new environmental conditions (normally, reference containing either artificial saliva or one of the salivary samples with various ions eluted from dental materials to be tested) and for an additional time interval of a further 48 h. At the end of the interval, the 72-h-old and 48-h-treated cultures were emptied of the various growth medium variants and washed with TFS.

To avoid the membrane and lytic attack of this proteolytic enzyme, the cell layer was rapidly detached from the wall of the culture flasks by rapid trypsinization. The detached cells were immediately resuspended in a growth medium with 2% fetal serum to inactivate the enzyme, resulting in cell suspensions. These have been used to obtain dilutions and subjected to flow cytometry to estimate the density of both cellular and early apoptosis (the separation of apoptosis from pre-apoptosis is conditioned by the binding of 7-AAD to DNA and annexin-5 FITC to the membrane, which ensures double signal for the two fluorochromes in apoptosis and single signal for annexin in the case of pre-apoptosis), as well as the number of living and dead cells.

The Beckman Cell Lab Quanta SC flow cytometer and two fluorochromes were used for the determination: 7-AAD (7-aminoactinomycin D). This method has been completed and presented in another publication [10].

## 3. Results

The microscopic structural changes observed at the level of the materials fall into the category of abrasions, perforations, as well as superficial defects (phenomena that are noticed mainly in the conditions of the oral environment at older restorations).

Remarkable changes are noticeable on the Ni-Cr alloy’s surface, with obvious matting and discoloration (Figure 1).

In Figure 2, the metallographic examination shows a facet/area of corrosion resulting from the ion release process.

Figure 3a,b shows the metal surface of the Cr-Co and Ni-Cr alloy affected by ion loss. The presence of corrosion cells at both surfaces was observed, with a tendency to converge in a larger corrosive zone, affecting the whole mass of material (in the case of Cr-Co alloy) and with tears of material for the Ni-Cr alloy (both against the background of pre-existing technological defects).

Figure 4 depicts the metallographic study of the Ni-Cr alloy surface quality prior to (a) and following (b) the onset of the ion loss phenomenon. After the corrosive attack (b), if the surface was initially smooth with only a few traces left over from the finishing process (a), pitting areas are discovered in the mass of material because of the release of nickel and chromium ions in the salivary environment, along with other components.

The decrease in the alloy’s stress resistance caused by the loss of constituent elements during the ion release process provides justification for the appearance of these fissure variations in the mass of the material [10,11]. For the pressure created by masticatory strains, for example, when an external energy input is applied (an input that could cause a crack to form), changes in the structure of the material (cracking, tearing of the material) are initiated and amplified by the conditions of the oral environment (pH, masticatory forces, masticatory cycle, occlusal relief) (Figure 5 and Figure 6).

Figure 7 shows the surface of the Cr-Co alloy that has not been processed to an advanced fineness due to the complex shape of the sample. However, after the electrochemical treatment, the appearance of corrosion points was found (Figure 7c,d), thus confirming the conclusions of other studies from the literature [12,13,14].

Figure 8 shows the surface modification of Ni-Cr. It was found that the corrosion process develops over the entire surface, starting from the boundaries of the dendrites in the microstructure of the alloy. The attack on the alloy is deep, and the corrosion points were large and deep, although the electrode potential and the treatment time were the same [15,16].

The surface composition of the alloys before and after the electrochemical treatment, analyzed by EDX technique, is shown in Figure 9 and Figure 10. The EDX spectra obtained for the surface of the alloys before corrosion, on the initial surface composition, and the surface composition after corrosion are highlighted. The corrosion process in the case of the Cr-Co alloy is characterized by shallow corrosion patches, superficial corrosion, and a small amount of dissolved metal. Nickel passed mostly in solution but in small quantities (≈2.5% of the amount of Ni in the alloy), unlike Chromium and Cobalt, which remain at this level (Figure 9).

Nickel (about 9% of the existing amount), Manganese, and Cobalt dissolve in the Ni-Cr alloy (Figure 10). In the case of the electrochemically treated sample, some of the Cobalt oxidizes on the alloy’s surface, converting to CuO (insoluble and adhering to the surface), which explains the presence of oxygen in the EDX spectrum.

The average values of the evaluation index of cellular protein synthesis reactivity are inserted in Table 1 and represented graphically in Figure 11. Table 1 shows the intensification of cellular protein synthesis because of average values of total protein concentration at 72 h, which gradually increased. In the Table 1, it can be seen that the evolution of control monkey kidney cell cultures is characterized, at the interval of 72 h, by average values of the concentration of total proteins, which, through their progressive increase, argues for the intensification of cellular protein synthesis, correlated (as we will see in the results obtained under the conditions of the investigation of the cell proliferation process) with the increase in the number of cells by amplifying the rate of cell multiplication, the factors inherently imprinting the normal development of the control cultures. Cell cultures treated with artificial saliva at 72 h show a significantly decreased average protein content. The influence of the protein synthesis disturbance is what causes the RM culture to be inhibited to the extent of 24.12%, the cell cultures treated with SA-Cr-Co ionic eluate to be inhibited to the extent of 34.41%, and the cell cultures treated with SA-Ni-Cr ionic eluate to be inhibited to the extent of 34.12% (Figure 11).

The degree of cell development was reduced compared to the control group, with values between 34.12% and 40%, when the ionic diffuses from the incubation of the sample made of Ni-Cr alloy and Cr-Co alloy were present in the culture media renal cells. Excluding the influence of artificial saliva as a vehicle of the released ions, the degree of cell growth inhibition by the dental materials alone was 10% (Ni-Cr alloy) and 10.29% for Cr-Co alloy.

Determining how varying ionic elutes of dental materials in artificial saliva affected the proliferation process of normal monkey kidney cells was another element. Compared to point 0, that of the initiation of cell cultures, we find, in the case of the control batch, a decrease in fluorescence intensity, which signals a normal division process. The decrease in this fluorescence is not very large, which can be explained by the fact that, when using normal cells without changes in the control of cell proliferation, the rate of cell division decreases when reaching the monolayer phase, a natural consequence of the functioning of contact inhibition. When monkey renal normal cell cultures were treated with SA-Cr-Co alloy, the registered profile overlaid the original one, and a negligible effect on the cell proliferation process was also observed (Figure 12a).

In the case of SA-Ni-Cr alloy, minor alterations in the proliferative profile of monkey kidney cell cultures were seen after treatment with the tested dental materials. However, these changes from the specific profile of the control and reference group (whose magnitude does not reach significant values) can be based on typical physiological variances in the process of cell proliferation. Voluntary contamination of incubated culture media with artificial saliva shows an increase in the viability of cell cultures (a decrease in the number of dead cells by 2.28%) (Figure 12b).

Thus, in the case of ionic incubation of the SA-Cr-Co alloy, we witness a decrease, but not significant one, in the number of living cells (88.5%), a corresponding increase in the number of dead cells (11.5%), and a negligible increase, of very low aptitude, of pre-apoptotic (0.006%) and apoptotic (0.037%) cells. Cr-Co alloy can be characterized as possessing a moderate cytotoxic potential expressed in cellular protein synthesis, cell viability, cell apoptosis, and the development of monkey kidney cell cultures used as biological material in our experiments (Figure 13a).

In the case of the interaction between Ni-Cr alloy and artificial saliva, much more aggressive action on the cells is observed, both biochemically and flow cytometrically. We witness a decrease, but not significant, in the number of living cells (87%), a corresponding increase in the number of dead cells (13%), and a negligible increase, of very low aptitude, of pre-apoptotic (0.006%) and apoptotic (0.048%) cells. The results recorded, as well as those on the reactivity of protein synthesis, draw attention to these samples, which appear to be endowed with a significant aggressive action on cellular cyto-physiological processes (Figure 13b).

## 4. Discussion

The surface qualities analyzed from the physical point of view by microscopic studies show that the variability of the images can be influenced not only by the quality of the material but also by its structural components and the process of release in artificial saliva of some components.

Often, the surface microscopic defects noticed in the form of depressions, gaps, cracks, ridges, or protrusions can constitute ecological niches for the bacterial flora. At the same time, they may represent local mechanical irritation factors for neighboring gingival tissues (microlesions) and starting points for irreversible macroscopic changes (corrosion) [17,18,19].

The results of the study show that the surfaces of metallic restorations bathed in saliva represent regions with different potentials from the rest of the surfaces. On the metallic surfaces observed, bacterial enzymes and other catabolism products are released, such as sulfides, chlorides, sulfonates, sulfur dioxide, and hydrogen sulfide, which will have an unfavorable effect on both the surface condition of metal prosthetic works and oral structures. Furthermore, added ions or other products released from the structure of the material under the action of oral factors may have a toxic or allergic effect on living structures [18,19,20].

Superficial defects (pores, cracks, holes or recesses, sulfides, etc.) represent areas of differentiated aeration that allow organic retention and reduced salivary circulation, which makes the electrochemical potential of these areas different from the rest of the surface of the prosthetic work; in fact, this generates the formation of galvanic micro piles, which favors electrochemical corrosion [5,7,21].

Some authors (Brugirard, Nocolson) point out that bimetallism often also contributes to the various heterogeneities of the salivary alloy or electrolyte, thus completing the picture of electrochemical corrosion, a fact highlighted in the study and often noted in clinical trials [1,3,6].

The corrosion process involves internal stresses resulting from the melting and casting of metal alloys which leads to stress corrosion and even intercrystalline and transcrystalline cracks; this reduces the reliability of the prosthetic work and results in subjective and objective changes, both prosthetically and in oral structures.

The results of existing studies in the literature show that the Co-Cr/Ni-Cr alloy can cause decreased cell proliferation, DNA damage, and apoptosis through released ions [22,23].

According to Jayaprakash et al., the percentage of ions released from the Ni-Cr alloy rises in direct proportion to the number of reforms [20]. Pouring 65% nickel-containing alloys dramatically raises cytotoxic activity, according to Imirzalioglu et al. [21]. Bearden and Cooke note that modest amounts of released Co ions hinder fibroblast growth, whereas greater concentrations result in morphological alterations and slowed fibroblast growth [24]. Akbar et al. mention that increased amounts of Cr and Co ions discharged into the oral environment dramatically reduce the viability of peripheral lymphocytes [25]. According to Bauer et al., the amount of Co and Cr ions released increases in direct proportion to the amount of chondrocyte apoptosis [23].

It is yet unknown how precisely Co ions have cytotoxic effects. The activation of inducible hypoxia factor (HIF) genes by Co ions might result in the creation of a hypoxic milieu, which in turn triggers the production of certain proinflammatory cytokines [22,23,24].

However, the corrosion process is amplified in the conditions of the oral environment, where the factors involved are numerous. Furthermore, it has multiple consequences on the restorative material; on the one hand, it decreases the restoration’s thickness per section, but on the other, it causes internal tensions due to mechanical pressures. This is the necessary framework for initiating structural changes, which happens through the released corrosion products (i.e., ions), and apparition of local inflammatory phenomena. Our study shows the same thing: local factors caused the electro-chemo-corrosion process to start, which in turn releases ions into the oral environment and has a negative impact on both the restoration’s surrounding tissues and the restoration itself.

The modifications in metal alloys for dental usage are caused by the primary alloy’s crystalline structure being disorganized throughout technical processes, particularly the casting and cooling of the alloy. Major segregation is the result of the higher specific gravity crystalline elements (such as iron or molybdenum) being positioned deeper within the alloy during cooling, while the lower specific gravity crystalline elements (such as Ag and Cu) are positioned on the alloy’s surface. This increases the surface area of the Ag and Cu crystals, making them more susceptible to a hostile oral environment. This encourages the discharge of highly concentrated ions, which also affects the material’s degree of hardness [4,11,20,26].

Common alloys are prone to corrosion as they lack structural features that would shield the alloy from corrosion agents; when technological processing flaws are included, it explains why corrosion cannot be prevented.

Structural modifications of the alloy, such as the inclusion (in its mass) of oxides, car-bides, and sulfides or the production of cracks and pores, reduce the resistance of the alloy to mechanical stress and increase the risk of electrochemical and stress corrosion.

## 5. Conclusions

In the case of the Cr-Co alloy, the corrosion occurs superficially, while in the case of the Cr-Co alloy, the corrosion defects are much larger and deeper, despite the electrode potential and similar treatment.

The analysis of the cell proliferation process of normal monkey kidney cell cultures subjected to the action of different dental materials does not suggest the existence of a perceptible impairment. These were well tolerated by the cells used in this experimental model and, we believe, by the cells from the composition of the oral tissues.

The Cr-Co and Ni-Cr alloy parts analyzed in artificial saliva behave as agents with aggressive action on cells, results obtained both by biochemical and flow cytometric testing. Dental materials based on Cr-Co and Ni-Cr alloys can be characterized as possessing a moderately expressed cytotoxic potential on cellular protein synthesis, cellular viability, and cellular apoptosis.

## Figures and Tables

**Figure 1 medicina-58-01524-f001:**
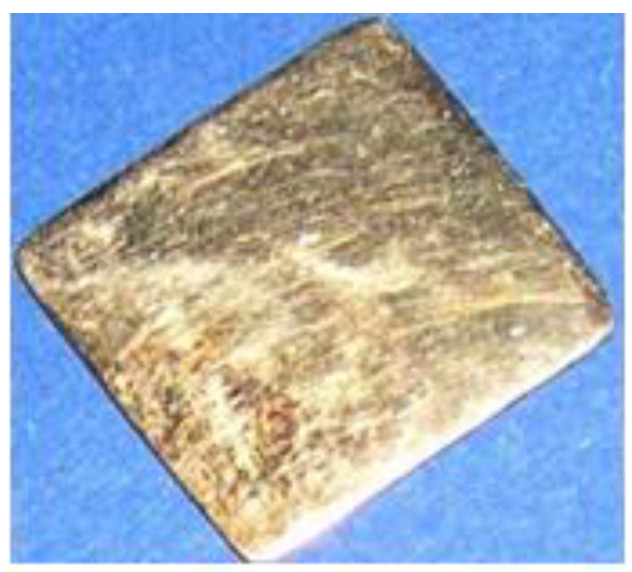
Ni-Cr alloy mass cracks.

**Figure 2 medicina-58-01524-f002:**
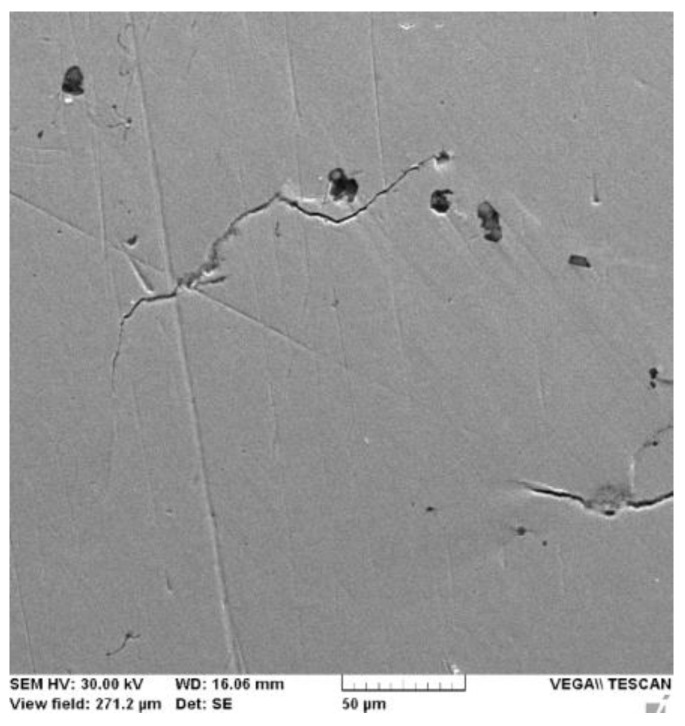
Ni-Cr surface corrosion cells. (WD—working distance—the distance between sample surface and the lower end of the pole piece where the electrons are coming from; HV—accelerated energy; SE—detector).

**Figure 3 medicina-58-01524-f003:**
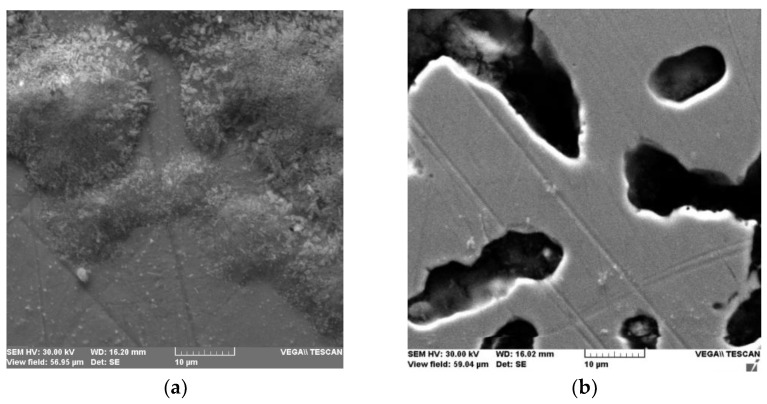
(**a**) Surface appearance of corroded Cr-Co alloy; (**b**) Ni-Cr skeleton affected by corrosion. (WD—working distance—the distance between sample surface and the lower end of the pole piece where the electrons are coming from; HV—accelerated energy; SE—detector).

**Figure 4 medicina-58-01524-f004:**
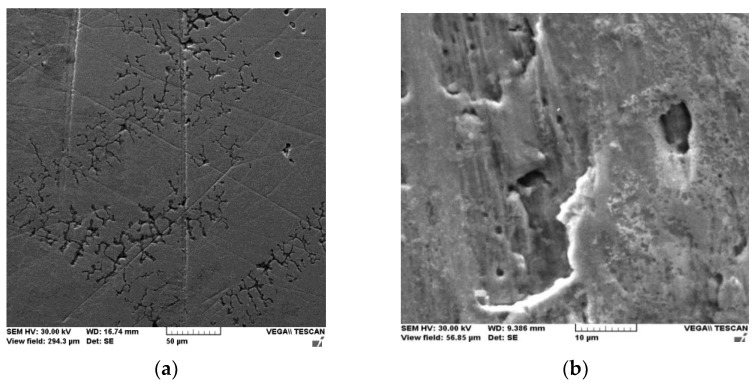
Metallographic appearance of the Ni-Cr surface before (**a**) and after (**b**) the occurrence of the corrosion process. (WD—working distance—the distance between sample surface and the lower end of the pole piece where the electrons are coming from; HV—accelerated energy; SE—detector).

**Figure 5 medicina-58-01524-f005:**
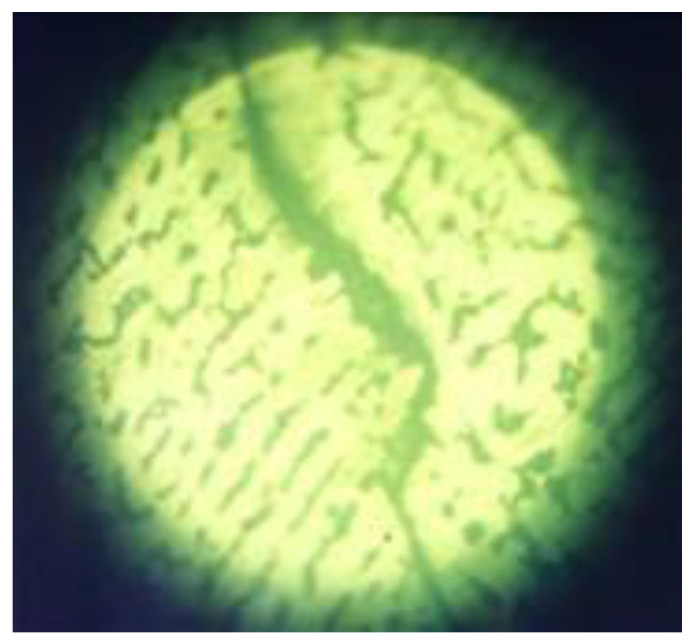
Intercrystalline crack.

**Figure 6 medicina-58-01524-f006:**
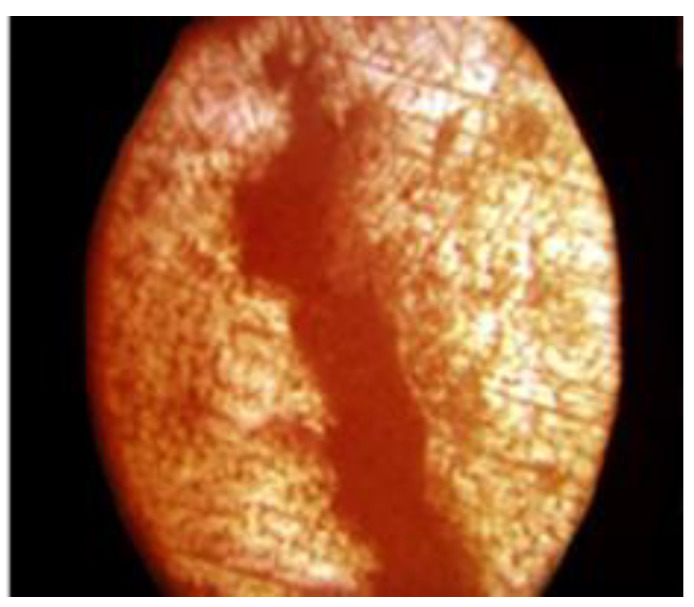
Cracking corrosion.

**Figure 7 medicina-58-01524-f007:**
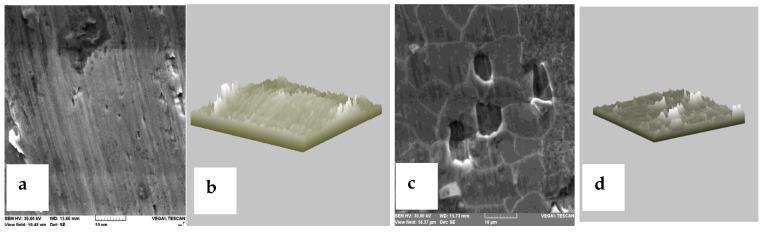
Microstructure of Cr-Co alloy surface before corrosion ((**a**)—the surface of the sample is not perfectly smooth; (**b**)—roughness is observed at the edges of the sample) and after corrosion ((**c**)—corrosion points on the surface of the sample; (**d**)—roughness distributed over the entire surface of the sample).

**Figure 8 medicina-58-01524-f008:**
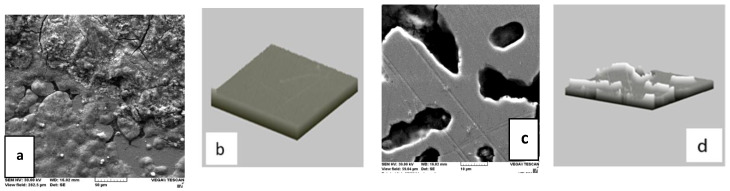
Microstructure of the Cr-Ni alloy surface before corrosion ((**a**)—cracks on the surface of the sample; (**b**)—the surface of the sample is uneven with small roughnesses of small sizes) and after corrosion ((**c**)—corrosion process develops over the entire surface; (**d**)—roughnesses of variables distributed over the entire surface of the sample).

**Figure 9 medicina-58-01524-f009:**
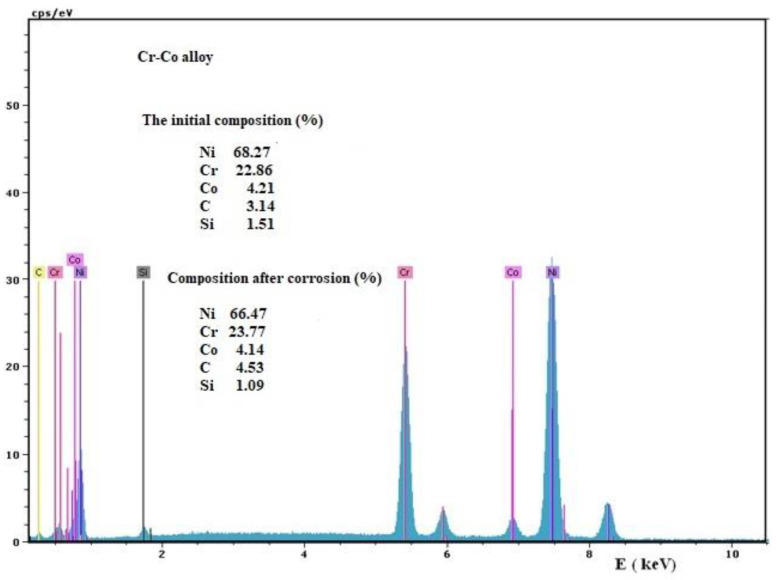
EDX spectrum of the Cr-Co sample before corrosion and surface compositions, initially and after corrosion.

**Figure 10 medicina-58-01524-f010:**
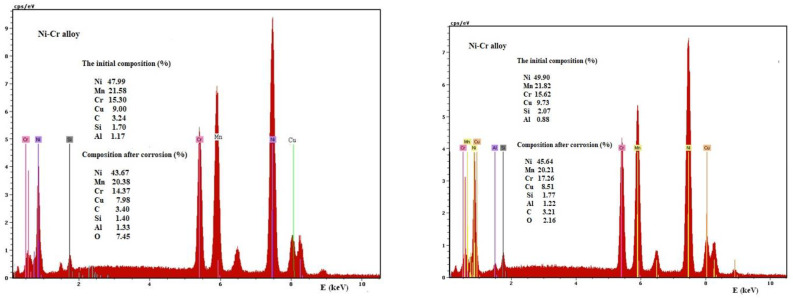
EDX spectrum of the Ni-Cr sample before corrosion and surface compositions, initially and after corrosion for two samples Ni-Cr alloy.

**Figure 11 medicina-58-01524-f011:**
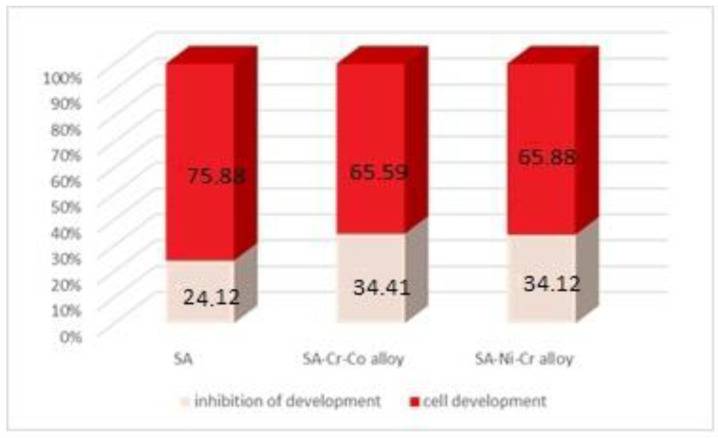
Alteration of the developmental process of monkey kidney cell cultures.

**Figure 12 medicina-58-01524-f012:**
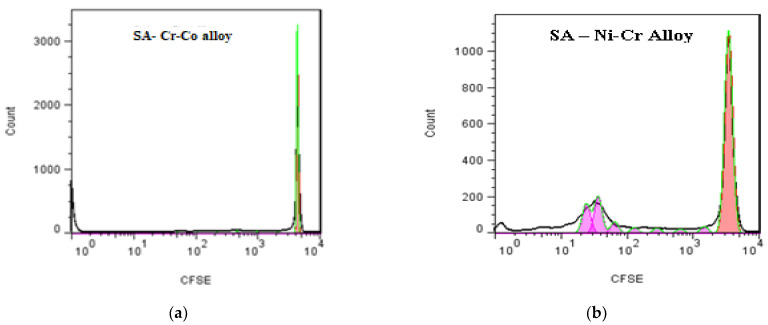
The process of cell proliferation of renal cell cultures, recorded by flow cytometer with blue laser (488 nm), under the conditions of their treatment with ionic eluate, from the incubation Cr-Co alloy (**a**) and Ni-Cr (**b**) with artificial saliva.

**Figure 13 medicina-58-01524-f013:**
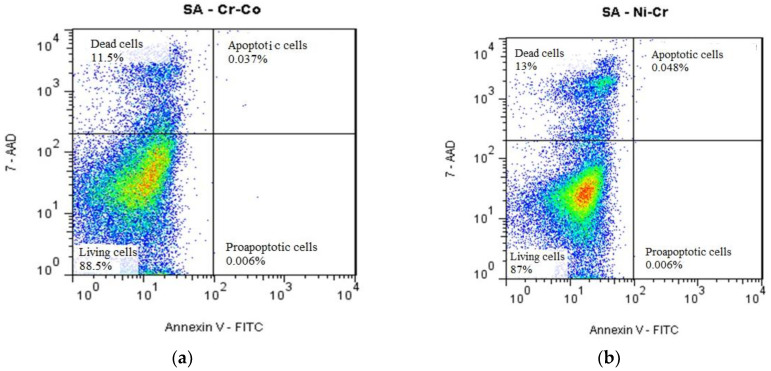
The process of apoptosis and cell viability in monkey renal cell cultures, recorded in continuous flow of cells at the flow cytometer, under the conditions of their treatment with artificial saliva and ionic elutes, from incubation of Cr-Co (**a**) and Ni-Cr (**b**) with artificial saliva.

**Table 1 medicina-58-01524-t001:** Ionic diffuses made from various dental materials boost the protein concentration (µg protein/culture) of monkey renal cell cultures that have been grown for 72 h.

Variant Cultures	Protein Concentration X ± ES	*p*
Control group	281.22 ± 6.38 (5)	-
SA-Ni-Cr alloy	185.27 ± 5.56 (5)	<0.01
SA-Ni-Cr alloy	179.42 ± 2.84 (5)	<0.001
SA-Cr-Co alloy	184.46 ± 4.09 (5)	<0.001

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
