# Peer review of "Analytical Study Regarding the Behavior of Cr-Co and Ni-Cr in Saliva"

_medicina, 2022, doi:10.3390/medicina58111524_

Round 1

Reviewer 1 Report

The manuscript investigates the in vitro interaction between dental alloys and artificial saliva, mainly focusing on the effect of saliva on alloy corrosion as well as changes in saliva composition. Some points have been raised and are described below:

1.  The English grammar needs improvement.

2. Too many errors in capitalization of letters, writing format, text layout, punctuation and other details in line24\ line112\ line270\ line279\ line287\ line312\ line313\line499\ line502.

3.  The relative contents of the Cr-Co alloy and Ni-Cr alloy have not been illustrated, which has a great effect on the corrosion resistance. The result of EDX only represents the compositions of the alloy in single points instead of the whole surface.

4.     Inconsistent naming of experimental groups confuses readers. The experimental group “Sa-Cr-Co alloy”in line 149 became “Sa-C alloy” in line 286. The group “Sa-Ni-Cr alloy”in line 149 became “SA- Ni-Cr-Co alloy”line 298.

5.     Why the kidney cells were chosen instead of the oral-related cells like epithelial cells or fibroblast given that the article is on the topic of restorative materials?

6.     The scale of figure 2, figure 3 and figure 4 is missing.

7.    Figure 2 and figure 3 (a) are identical except for the hues, length-to-width ratio and annotations. Figure 2 shows “Cr-Ni alloy surface” and Figure 3 shows “Cr-Co alloy surface”.

8.    Lacking clear and precise explanation in annotations makes it hard to understand what the two subfigures in Figure 10 try to illustrate.

9.    Too little information is presented in tables and figures, making statements in the text groundless. For example, the statement “protein concentration gradually increase” has not been demonstrated in the Table II.

10. The DISCUSSION is repetitive, lengthy and tedious. It should focus on the expansion knowledge based on your experimental results instead of going over the causes and influences of electrochemical corrosion again, which have already been fully illustrated in INTRODUCTION.

11. The purpose of the CONCLUSION is to present the results of your experiment in a simple and visual form, rather than repeat the necessity of the experiment again.

Author Response

Response to Reviewer 1 Comments

  1. The English grammar needs improvement.

R: The material was read and corrected by a native English speaker.

  1. Too many errors in the capitalization of letters, writing format, text layout, punctuation, and other details in line24\ line112\ line270\ line279\ line287\ line312\ line313\line499\ line502.

R: I tried to correct all the inadvertences pointed out by you.

  1. The relative contents of the Cr-Co alloy and Ni-Cr alloy have not been illustrated, which has a great effect on the corrosion resistance. The result of EDX only represents the compositions of the alloy in single points instead of the whole surface.

R: I have completed in the Material and methods section the composition of the materials used in the study (row 109-110).

  1. Inconsistent naming of experimental groups confuses readers. The experimental group “Sa-Cr-Co alloy”in line 149 became “Sa-C alloy” in line 286. The group “Sa-Ni-Cr alloy”in line 149 became “SA- Ni-Cr-Co alloy”line 298.

R: I have corrected the name of the materials in the text.

  1. Why the kidney cells were chosen instead of the oral-related cells like epithelial cells or fibroblast given that the article is on the topic of restorative materials?

R: This study is part of a larger research, being chosen from the beginning the cultural environment. Monkey kidney cell cultures were chosen because this culture medium exhibits some degree of contact inhibition after monolayer formation and is therefore useful in growing slowly replicating cells. It also ensures very good viability of control cells, argued by a major weight of living cells and the presence of a very low percentage of dead cells.

  1. The scale of figure 2, figure 3 and figure 4 is missing.

R: I have replaced the images that do not show scales.

  1. Figure 2 and figure 3 (a) are identical except for the hues, length-to-width ratio and annotations. Figure 2 shows “Cr-Ni alloy surface” and Figure 3 shows “Cr-Co alloy surface”.

R: I have replaced the indicated images with more suggestive ones.

  1. Lacking clear and precise explanations in annotations makes it hard to understand what the two subfigures in Figure 10 try to illustrate.

R: I have corrected and completed the annotations of figures 10a and 10b.

  1. Too little information is presented in tables and figures, making statements in the text groundless. For example, the statement “protein concentration gradually increase” has not been demonstrated in the Table I.

R: I have added additional information regarding the results from Table I (rows 280-288).

  1. The DISCUSSION is repetitive, lengthy, and tedious. It should focus on the expansion of knowledge based on your experimental results instead of going over the causes and influences of electrochemical corrosion again, which have already been fully illustrated in

R: I removed the paragraphs that were not suggestive from the "Discussions" section.

  1. The purpose of the CONCLUSION is to present the results of your experiment in a simple and visual form, rather than repeat the necessity of the experiment again.

R: At your suggestion, we reformulated the conclusions of the study.

Reviewer 2 Report

Dear Authors,

The conducted research is carried out in accordance with a good research plan.

Unfortunately, the article has an incorrect name. Unfortunately, you do not compare dental materials, but only one dental material (Co-Cr) with Cr-Ni, which is not used in any application known to me today.

There are comparisons with titanium materials or other Cr alloys in your literature. Comparing with a nickel alloy is a misunderstanding. Nickel is a material considered harmful and has been withdrawn from use in dental alloys.

Therefore, the title should certainly be redrafted. This is not a review article and only contains a small excerpt from the problem presented in the title.

However, the subject matter presented in the article is already described in such a way that there is basically no reason to raise it again. Why should we find out again that nickel is bad and releases into saliva? this is obvious. Just like the fact that it sensitizes patients in a significant percentage of the population. for this reason, such alloys have not been used for at least a dozen or so years.

I also have fundamental reservations about the quality of SEM photos. In fact, nothing can be said about them because they are in low resolution. I have the impression that the samples are scratched but it is impossible to judge because the resolution is too low. The assessments of visible changes are also, I think, a bit exaggerated.

In my opinion, the materials for this article were wrongly selected.

After editing the introduction and potential uses and changing the title, the article may be able to be published. It is also necessary to improve the quality of SEM photos so that you can even judge what you see in them.

Author Response

Response to Reviewer 2 Comments

  1. Unfortunately, the article has an incorrect name. Unfortunately, you do not compare dental materials, but only one dental material (Co-Cr) with Cr-Ni, which is not used in any application known to me today.

R: At your suggestion, I changed the title of the article: "Analytical study regarding the behavior of Cr-Co and Cr-Ni in saliva" instead of "Comparative study on the interaction between dental materials and saliva"

2, There are comparisons with titanium materials or other Cr alloys in your literature. Comparing it with a nickel alloy is a misunderstanding. Nickel is a material considered harmful and has been withdrawn from use in dental alloys.

R: I deleted the paragraph and the bibliographic reference from the text.

  1. Therefore, the title should certainly be redrafted. This is not a review article and only contains a small excerpt from the problem presented in the title.

R: I changed the title because we only performed an analysis of the behavior of the 2 alloys in saliva.

  1. However, the subject matter presented in the article is already described in such a way that there is basically no reason to raise it again. Why should we find out again that nickel is bad and releases into saliva? this is obvious. Just like the fact that it sensitizes patients in a significant percentage of the population. for this reason, such alloys have not been used for at least a dozen or so years.

R: Although there are studies that demonstrate the behavior of alloys with Ni, we tried to determine by flow cytometry the influence of Ni, Cr and Co ions on the viability of the cells in the culture medium. The Cr-Ni alloy is still used in Romania as a metal support for the fixed prosthesis.

  1. I also have fundamental reservations about the quality of SEM photos. In fact, nothing can be said about them because they are in low resolution. I have the impression that the samples are scratched but it is impossible to judge because the resolution is too low. The assessments of visible changes are also, I think, a bit exaggerated.

R: : I have changed the SEM images with others more suitable for the intended purpose.

  1. In my opinion, the materials for this article were wrongly selected.

R: This study is part of a larger research in which we analyzed the behavior of several materials used in Romania. I chose these materials because they are frequently used in prosthetics in our country.

Round 2

Reviewer 1 Report

none